# Maresin 1 Exerts a Tissue-Specific Regulation of Adipo-Hepato-Myokines in Diet-Induced Obese Mice and Modulates Adipokine Expression in Cultured Human Adipocytes in Basal and Inflammatory Conditions

**DOI:** 10.3390/biom13060919

**Published:** 2023-05-31

**Authors:** Leyre Martínez-Fernández, Miguel Burgos, Neira Sáinz, Laura M. Laiglesia, José Miguel Arbones-Mainar, Pedro González-Muniesa, María J. Moreno-Aliaga

**Affiliations:** 1Department of Nutrition, Food Science and Physiology, Center for Nutrition Research, School of Pharmacy and Nutrition, University of Navarra, 31008 Pamplona, Spain; 2IDISNA-Navarra Institute for Health Research, 31008 Pamplona, Spain; 3Adipocyte and Fat Biology Laboratory (AdipoFat), Unidad de Investigación Traslacional, Instituto Aragonés de Ciencias de la Salud (IACS), Instituto de Investigación Sanitaria (IIS) Aragón, Hospital Universitario Miguel Servet, 50009 Zaragoza, Spain; 4CIBER Fisiopatología de la Obesidad y Nutrición (CIBEROBN), Instituto de Salud Carlos III (ISCIII), 28029 Madrid, Spain

**Keywords:** myokines, hepatokines, Maresin 1, specialized pro-resolving mediators, inflammation

## Abstract

This study analyses the effects of Maresin 1 (MaR1), a docosahexaenoic acid (DHA)-derived specialized proresolving lipid mediator with anti-inflammatory and insulin-sensitizing actions, on the expression of adipokines, including adiponectin, leptin, dipeptidyl peptidase 4 (DPP-4), cardiotrophin-1 (CT-1), and irisin (FNDC5), both in vitro and in in vivo models of obesity. The in vivo effects of MaR1 (50 μg/kg, 10 days, oral gavage) were evaluated in epididymal adipose tissue (eWAT), liver and muscle of diet-induced obese (DIO) mice. Moreover, two models of human differentiated primary adipocytes were incubated with MaR1 (1 and 10 nM, 24 h) or with a combination of tumor necrosis factor-α (TNF-α, 100 ng/mL) and MaR1 (1–200 nM, 24 h) and the expression and secretion of adipokines were measured in both models. MaR1-treated DIO mice exhibited an increased expression of *adiponectin* and *Ct-1* in eWAT, increased expression of *Fndc5* and *Ct-1* in muscle and a decreased expression of hepatic *Dpp-4*. In human differentiated adipocytes, MaR1 increased the expression of *ADIPONECTIN*, *LEPTIN*, *DPP4*, *CT-1* and *FNDC5*. Moreover, MaR1 counteracted the downregulation of *ADIPONECTIN* and the upregulation of *DPP-4* and *LEPTIN* observed in adipocytes treated with TNF-α. Differential effects for TNF-α and MaR1 on the expression of *CT-1* and *FNDC5* were observed between both models of human adipocytes. In conclusion, MaR1 reverses the expression of specific adipomyokines and hepatokines altered in obese mice in a tissue-dependent manner. Moreover, MaR1 regulates the basal expression of adipokines in human adipocytes and counteracts the alterations of adipokines expression induced by TNF-α in vitro. These actions could contribute to the metabolic benefits of this lipid mediator.

## 1. Introduction

Obesity is a global challenge since its prevalence continues growing, reaching epidemic proportions [1]. This disease is characterized by an abnormal fat accumulation accompanied by a low-grade chronic inflammation, which is in turn associated with the development of metabolic disorders including insulin resistance, type 2 diabetes mellitus (T2DM) and cardiovascular disease [2,3]. White adipose tissue (WAT) plays a key role in the pathogenesis of obesity and associated complications. In addition to its primary role as an energy storage organ, WAT has been revealed as a relevant endocrine organ, which releases a wide range of soluble bioactive molecules including hormones, adipokines, cytokines, chemokines, enzymes, lipids, mRNAs and microRNAs with key impact on insulin sensitivity and metabolism, as well as systemic immune response [4,5,6]. In obesity, hyperplasic and hypertrophied WAT produces an altered secretome, which through its autocrine and paracrine actions in WAT and its endocrine effects, in liver and muscle among other tissues, causes an altered metabolic state with insulin resistance [7].

In the last decades, it has been discovered that n-3 polyunsaturated fatty acids (n-3 PUFAs) can be enzymatically converted into potent anti-inflammatory and pro-resolving lipid mediators called specialized pro-resolving lipid mediators (SPMs) [8,9]. These SPMs act as braking-signals of inflammatory response as well as facilitators of timely resolution of inflammation at nanomolar and even picomolar range [10]. Some of these SPMs include maresins (MaR), resolvins (RvD) and protectins (PD) derived from n-3 PUFAs, or lipoxin (LX) A4, derived from n-6 PUFAs. Maresin 1 (MaR1) is a docosahexaenoic acid (DHA)-derived lipid mediator with potent anti-inflammatory and pro-resolving properties [11] Moreover, MaR1 exerts beneficial metabolic effects including insulin sensitizing actions [12] and liver protection against fatty acid lipotoxicity and endoplasmic reticulum stress [13,14,15]. A previous study by our group has revealed MaR1 capacity to regulate adipocytokines’ expression towards an anti-inflammatory profile in both DIO and genetically-induced obese mice [12]. These data suggest that MaR1 systemic actions could be mediated by the regulation of adipokines production. In the current study we have focused on adipokines with an important role in glucose and lipid metabolism including adiponectin, leptin, DPP-4, CT-1 and irisin [16,17,18,19,20]. Leptin, which is essential for the central regulation of food intake and maintenance of body weight, also exerts relevant physiological functions in peripheral tissues associated with glucose and lipid metabolism, including liver and skeletal muscle [17]. Another well-studied hormone is adiponectin which plays a crucial role in glucose and lipid metabolism, and protects against inflammation and oxidative stress [19]. Lower adiponectin levels have been reported in obese patients with metabolic syndrome [19], as well as in prediabetic states [21]. Previous studies by our group and others revealed that cardiotrophin 1 (CT-1) is a key regulator of energy homeostasis, along with glucose and lipid metabolism [18,22]. The current study has also focused on DPP-4, which was identified in 2011 as a novel adipokine that can impair insulin sensitivity in an autocrine and paracrine fashion [16]. Interestingly, inhibitors of DPP-4 have been recently employed as a new class of antidiabetic agents [23]. Finally, irisin is another adipokine/myokine, induced by exercise in mice and humans, that has been proposed to induce “browning” of WAT, promoting thermogenesis and energy expenditure [20].

To our knowledge, little is known regarding MaR1 tissue-specific regulation of these metabolic relevant bioactive molecules. On the other hand, previous studies by our group have shown that MaR1 ameliorates TNF-α-induced alterations on insulin-stimulated glucose uptake [24] and on lipolysis and autophagy in adipocytes [25]. However, the ability of MaR1 to counteract the alterations induced by this cytokine on the production of adipokines has not been reported yet.

Therefore, the aim of the present study was to analyze the efficacy of MaR1 to reverse changes induced by obesity on the expression of these adipokines in WAT, liver and muscle of DIO mice. Moreover, we studied the direct effects of MaR1 on basal and TNF-α altered production of adipokines in two models of primary cultured human adipocytes.

## 2. Materials and Methods

### 2.1. Animal Model

Seven-week-old C57BL/6J male mice (Harlan Laboratories, Spain) were housed under controlled conditions (22 ± 2 °C, with a 12 h light–dark cycle, relative humidity, 55 ± 10%). Animals were fed with a standard mouse pelleted chow diet (13% of kcal from fat, 67% from carbohydrates and 20% from protein) from Harlan Teklad Global Diets (2014 diet, Harlan Laboratories, Indianapolis, IN, USA) for an adaptation period of 7 days. Then, one group (control, n = 7) was fed with standard mouse chow diet, and a second group (DIO, n = 16) with a high-fat diet (HFD, 60% of kcal from fat, 20% from carbohydrates and 20% from protein) provided by Research Diets (New Brunswick, NJ, USA), for three months. Thereafter, the control group received daily oral gavage of the vehicle (100 μL of sterile saline—0.1% ethanol) for 10 days. Additionally, DIO mice were assigned into two sub-groups that received a daily oral gavage of the vehicle (*n* = 8) or MaR1 (50 μg/kg body weight, *n* = 8, Cayman Chemical, Ann Arbor, MI, USA) for 10 days. Body composition was measured at the end of the study by magnetic resonance technology (EchoMRI-100-700; Echo Medical Systems, Houston, TX, USA) in non-fasted mice.

Mice were sacrificed after overnight fasting, blood was extracted and epididymal white adipose tissue (eWAT), liver and muscle tissue (soleus) were immediately harvested, weighed and snap-frozen in liquid nitrogen before being stored at −80 °C for further determinations, as previously described by our group [26]. All experimental procedures were performed according to Animal Care guidelines, and with the approval of the Ethics Committee for Animal Experimentation of the University of Navarra (Protocol number: 047-15) in accordance with the EU Directive 2010/63/EU.

### 2.2. Biochemical Analyses

Serum biochemical determinations in animal samples were carried out after a 16 h fasting period. Total cholesterol, triglycerides (TG) and alanine amino transferase (ALT) serum levels were determined using a Pentra C200 autoanalyzer (Roche Diagnostic, Basel, Switzerland), following the manufacturer’s instructions. Fasting glucose levels were measured with a standard glucometer (Accu-Chek Advantage; Roche Diagnostic).

### 2.3. Cell Culture and Differentiation of Human Subcutaneous Pre-Adipocytes

Two models of human subcutaneous adipocytes were used for the cell culture studies. First, human mesenchymal stem cells (hMSC) from subcutaneous abdominal adipose tissue of non-diabetic obese male (n = 1) and female (n = 2) donors (age: 32–77 years; BMI: 30–41 kg/m^2^) [27], obtained by Dr José Miguel Arbonés-Mainar (Instituto Aragonés de Ciencias de la Salud, Zaragoza, Spain). This study was approved by the Institutional Review Board (Research Ethic Committee of Aragón, CEICA 20/2014), and informed consent was obtained from all participants.The hMSC were maintained and differentiated as previously described [12,28]. Briefly, the hMSC were maintained in 10% FBS-low-glucose DMEM, 1 mM pyruvate, 4 mM, glutamine and 1% of antibiotics (10,000 Units/mL penicillin; 10,000 μg/mL streptomycin; Gibco, Thermo Fisher Scientific, Waltham, MA, USA) (37 °C, 5% CO_2_). Two days after confluence the hMSC were differentiated into adipocytes using an adipogenic cocktail for 6 days. The standard adipogenic cocktail consisted of 10% FBS–high-glucose DMEM plus 1.5 µM insulin, 1 µM dexamethasone, 500 µM 3-isobutyl-1-methylxanthine, and 1 µM rosiglitazone and 1% of antibiotics. Adipocytes were maintained in 10% FBS–high-glucose DMEM, 1 mM pyruvate and 4 mM glutamine and considered mature at 6–8 days post-differentiation.

Second, commercially available cryopreserved human subcutaneous preadipocytes (hSP) from different lots of non-diabetic overweight male and female donors pooled from nondiabetic overweight-obese male and female donors (BMI: 26.85–33.37 kg/m^2^) were purchased from Zen-Bio (Research Triangle Park, Durham, NC, USA) and differentiated according to Prieto et al. [29]. Briefly, cryopreserved pre-adipocytes were cultured in pre-adipocyte medium (PM-1; DMEM/Ham’s F-12 medium, HEPES, FBS, penicillin, streptomycin, amphotericin B; Zen-Bio) until confluence. To induce differentiation, PM-1 medium was replaced with 1 mL of differentiation medium (DM2; Zen-Bio) including biotin, pantothenate, human insulin, dexamethasone, isobutylmethylxanthine, and a peroxisome proliferator activated receptor gamma agonist (days 0–7). After 7 days, 600 µL of DM-2 medium was removed and 800 µL of adipocyte medium (AM1; Zen-Bio) containing PM-1, biotin, pantothenate, human insulin, and dexamethasone were added. Media were replaced by 800 µL of fresh AM1 every 48 h. By day 14 of incubation, cells contained large lipid droplets and were considered mature adipocytes. The study was approved by the Research Ethics Committee of the University of Navarra (032/2012).

### 2.4. In Vitro Treatments

Fully differentiated adipocytes (hMSC and hSP) were treated with MaR1 (Cayman Chemical; Ann Arbor, MI, USA) at concentrations between 1 and 10 nM or between 1 to 200 nM for those cultures that were additionally treated with TNF-α (100 ng/mL; Peprotech EC Ltd. London, UK) for 24 h, as previously described [30]. MaR1 and TNF-α were dissolved in ethanol or ultra-purified water, respectively, and the final volume of the treatments in each well was 0.1%. Control cells were treated under the same vehicle conditions. After 24 h of treatment, culture media were collected and cells were processed to isolate the corresponding total RNA, which were stored at −80 °C for subsequent analysis.

As previously described [25,31], cell viability was determined by lactate dehydrogenase (LDH) activity in the supernatant, an indicator of cell membrane integrity, and therefore, as a measurement of cell necrosis/apoptosis [31]. Adipocytes treated with MaR1, at all the concentrations tested, did not show differences with vehicle or with TNF-α, indicating that MaR1 did not alter cell viability.

### 2.5. Analysis of mRNA Expression by Real-Time PCR

Total RNA from eWAT, soleus muscle, liver and human cultured adipocytes was extracted with TRIzol^®^ (Invitrogen, Carlsbad, CA, USA) or QIAzol^®^ reagent (Qiagen; Venlo, Limburg, The Netherlands). RNA quality and concentrations were measured by Nanodrop Spectrophotometer ND1000 (Nanodrop Technologies, Inc. Wilmington, NC, USA). RNA (1–4 μg) was then incubated with DNase I (DNA-free kit; Ambion, Austin, TX, USA or RapidOut DNA Removal Kit, Molecular Biology, Thermo Fisher Scientific) for 30 min at 37 °C. Thereafter, RNA was reverse-transcribed to cDNA utilizing Moloney murine Leukemia virus reverse transcriptase (Invitrogen) or the High-Capacity cDNA reverse transcription kit (Applied Biosystems, Thermo Fischer Scientific) following the manufacturers’ instructions. Real-time PCR was performed using the ABI PRISM 7900HT Fast System Sequence Detection System (Applied Biosystems, Foster City, CA, USA) and the Touch Real-Time PCR System (C1000 + CFX384, BIO-RAD, Hercules, CA, USA). Mice *AdipoQ, Lep* and *Ct-1* as well as human *ADIPOQ, DPP-4* and *LEP* mRNA levels were determined using predesigned TaqMan^®^ Assays-on-Demand and TaqMan Universal Master Mix (Applied Biosystems). Human *CT-1* and *FNDC5* and mice *Fndc5* and *Dpp-4* expression were analyzed by Power SYBR Green PCR Master Mix (BIO-RAD). Primers used were: human *CT-1* (Fw: 5′- CACTTGGAGGCCAAGATCC-3′; Rv: 5′-TCTCCCTGGAGCTGCACAT-3′), human *FNDC5* (Fw: 5′- TGAGAAGATGGCCTCCAAGAAC-3′; Rv: 5′-AGAAGAGGGCAATGACACCTG-3′), mice *Fndc5* (Fw: 5′- GGTGCTGATCATTGTTGTGG-3′; Rv: 5′-CGCTCTTGGTTTTCTCCTTG-3′), mice *Dpp-4* (Fw: 5′- TTCTGGGACTGCTTGGTGTC-3′; Rv: 5′-GCCGCTTCATCTTTGCTCAG-3′).

Primers were either extracted from other publications [20,32] or designed with Primer-Blast software (National Center for Biotechnology Information, Bethesda, MD, USA; https://www.ncbi.nlm.nih.gov/tools/primer-blast/, accessed on 31 May 2017). The efficiency and specificity of each primer pair was calculated as previously described [12]. Relative expression was determined by the 2^−ΔΔCT^ method after internal normalization to 18S.

### 2.6. Determination of Adipokine Levels in the Culture Media

The concentration of secreted adiponectin, leptin, DPP-4, CT-1 and irisin to the culture media was determined after 24 h of treatment using human adiponectin, human leptin, human irisin (R&D Systems) human DPP-4 (Thermo Fisher Scientific Waltham, MA, USA) and CT-1 (DLdevelop) ELISA kits following the manufacturer’s instructions.

### 2.7. Statistical Analysis

Data are presented as mean ± standard error (SEM). Comparisons between the values for different variables were analyzed by one-way ANOVA followed by Bonferroni post hoc tests or by Student’s *t*-test or Mann-Whitney U-test once the normality had been screened using Kolmogorov–Smirnov and Shapiro–Wilk tests. Statistical analyses and graphs were carried out using GraphPad Prism 9 software (Graph-Pad Software Inc., San Diego, CA, USA). A *p* value < 0.05 was considered significant.

## 3. Results

### 3.1. Tissue-Dependent Effects of MaR1 on Adipokines Expression in DIO Mice

As expected, body weight, fat mass, fasting glucose, cholesterol and ALT levels were significantly increased in DIO mice as compared to control (Appendix A), although the level of triglycerides were unchanged. Interestingly, levels of fasting glucose were lower in DIO-MaR1-treated mice compared to DIO mice. These results are in line with previous findings by our group [15,26]. In addition, MaR1 also attenuated (*p* < 0.01) the increased serum ALT levels observed in DIO mice, suggesting an amelioration of liver steatosis, as previously reported [15].

We performed a comparative analysis of the expression of key adipokines in WAT and the key metabolic tissues (liver and muscle) in control and DIO mice with or without MaR1 treatment. As expected, adiponectin exhibited a higher expression in eWAT than in muscle and liver (Figure 1A). After three months of HFD feeding there was a marked decline in the expression of adiponectin in eWAT (*p* < 0.001), whereas it was upregulated in muscle tissue (*p* < 0.01) and tended to be increased in liver of DIO mice. Remarkably, MaR1-treated mice exhibited increased adiponectin expression in eWAT compared to DIO mice (*p* < 0.05). However, MaR1 did not significantly affect adiponectin levels in liver and muscle.

Moreover, *leptin* was significantly upregulated in DIO compared to control mice in eWAT and muscle (*p* < 0.05). However, *leptin* expression in MaR1-treated mice in both tissues remained unchanged in relation to DIO mice (Figure 1B). Compared with those in control animals, muscle expression of *Dpp-4*, *Ct-1* and *Fndc5* was decreased in DIO mice (*p* < 0.05), whereas MaR1 treatment significantly reversed (*p* < 0.05) these inhibitory effects of the HFD on *Ct-1* and *Fndc5* (Figure 1C–E). In addition, MaR1 was also able to increase *Ct-1* expression in eWAT (*p* < 0.01 vs. DIO mice and *p* < 0.05 vs. control mice). Furthermore, hepatic expression of *Dpp-4* was promoted by HFD feeding (*p* < 0.05) and MaR1 significantly counteracted this increase by almost reaching lean control mice expression levels (*p* < 0.05). Similar effects were observed for *Fndc5* in liver, but significant differences were not found.

### 3.2. MaR1 Regulates the Expression of Glucose Homeostasis-Related Adipokines in Cultured Human Adipocytes

To evaluate the effects of MaR1 on the regulation of key adipokines, two different models of human differentiated adipocytes were used: (i) the human mesenchymal stem cells (hMSC) from subcutaneous abdominal adipose tissue and (ii) the human subcutaneous preadipocytes from subcutaneous abdominal adipose tissue (hSP), both from overweight and obese subjects.

MaR1 significantly increased the mRNA expression of *ADIPOQ*, *DPP-4*, and *CT-1* in both hMSC and hSP-derived adipocytes (Figure 2A,B). On the other hand, MaR1 induced cell-model-specific effects on the expression of *LEP* and *FNDC5*. MaR1 significantly increased *LEP* and *FNDC5* mRNA levels in hMSC adipocytes, while this effect was more moderated, and only observed in hSP-derived adipocytes after 1 nM MaR1 treatment in *LEP* expression (Figure 2A,B).

Then, in order to test if the changes observed at gene expression level were correlated with changes in adipokine secretion, we analyzed the culture media by using different ELISAs after 24 h-treatment with MaR1 (1 and 10 nM) in both adipocyte models. Similar to what was observed on *ADIPOQ* mRNA expression, MaR1 treatment induced a moderate but statistically significant increase in adiponectin secretion in hMSC-derived adipocytes, in contrast to hSP-derived adipocytes (Appendix A, respectively). MaR1 treatment also promoted a trend to increase Leptin and DPP-4 secretion in hMSC-derived adipocytes after MaR1 treatment (Appendix A). However, this trend was not observed in hSP-derived adipocytes (Appendix A). Regarding irisin, no correlation was found between *FNDC5* mRNA expression and irisin secretion in both adipocytes’ models (Appendix A). Although CT-1 secretion was also analyzed, the results obtained did not reach the minimum detection limit of the ELISA.

### 3.3. MaR1 Counteracts the Alterations Induced by TNF-α on Adipokines Expression in Cultured Human Adipocytes

In light of the potent anti-inflammatory actions described for MaR1 on adipose tissue in obesity [12], we also evaluated the ability of this SPM to counteract the alterations induced by TNF-α on adipokines expression/secretion. For this purpose, a higher range of MaR1 concentrations were tested (1–200 nM) in both models of human adipocytes as an attempt to overcome the strong TNF-α pro-inflammatory actions (Figure 3 and Appendix A). As expected, TNF-α exposure for 24 h reduced *ADIPOQ* mRNA expression in both models of adipocytes. This reduction was counteracted by MaR1 treatment at all doses in hMSC-derived adipocytes and hSP-derived adipocytes (Figure 3A, left and right panels). On the other hand, treatment with TNF-α showed an increase in *LEP* mRNA levels in both models. Only the highest dose of MaR1 (200 nM) significantly reversed this effect in hMSC-derived adipocytes (Figure 3B, left panel), while this reduction was observed in hSP-derived adipocytes at 1, 10 and 100 nM (Figure 3B, right panel).

Conversely, *DPP-4* mRNA expression increased under TNF-α exposure, and treatment at 10, 100 and 200 nM of MaR1 in hMSC adipocytes and 200 nM in hSP-derived adipocytes significantly reversed this effect (Figure 3C, left and right panels, respectively). Regarding the expression of *CT-1* and *FNDC5* after treatments, different effects were observed comparing hMSC and hSP-derived adipocytes. No relevant effects were detected for TNF-α on the gene expression levels of *CT-1* and *FNDC5* in hMSC-derived adipocytes, but the higher doses of MaR1 increased the expression levels of both genes (Figure 3D,E, left panels). On the other hand, *CT-1* and *FNDC5* expression were increased in hSP-derived adipocytes after TNF-α treatment, and this effect was reversed at the highest dose of MaR1 (Figure 3D,E, right panels).

Regarding the effects of TNF-α alone or in combination with MaR1 on the secretion of adiponectin and leptin, our data show that their effects were much more moderate than those observed on secretion, and no statistical differences were reached for any treatment in both adipocyte models (Appendix A). DPP-4 secretion also tended to increase in TNF-α treated adipocytes, but no effect was observed in the presence of MaR1 (Appendix A). Finally, irisin secretion was not changed after treatments in hMSC-derived adipocytes, while it tended to increase in hSP-derived adipocytes after TNF- α addition, with no effect after MaR1 treatment (Appendix A). Although CT-1 secretion was also analyzed, the results obtained did not reach the minimum detection limit of the ELISA.

## 4. Discussion

To confirm the potent anti-inflammatory actions described for MaR1 on adipose tissue in obesity, we proved the ability of this SPM to regulate the adipokines’ expression in different animal tissues of DIO mice and to counteract the alterations induced by TNF-α in two models of cultured human adipocytes (hSP and hMSC) from overweight/obese subjects.

Our results prove the modulatory capacity of MaR1 on the expression of key adipokines involved in insulin sensitivity and glucose homeostasis. Previously, we described that MaR1 treatment was able to increase basal adiponectin expression and secretion in human adipocytes from overweight/obese subjects [12]. Our current findings indicate that MaR1 treatment was able to counteract the TNF-α lowering actions on *ADIPOQ* expression in cultured human adipocytes (hMSC and hSP) from overweight/obese subjects as well as in eWAT of DIO mice treated with MaR1. Adiponectin has been widely studied as a promising therapeutic tool for T2DM and metabolic syndrome management, as it exerts anti-diabetic, anti-inflammatory and anti-atherogenic actions [19]. Lower adiponectin expression and circulating levels are associated with dysfunctional adiposity and precede the deterioration of insulin sensitivity in the prediabetic state [21], thus predicting the development of insulin resistance and T2DM [33,34]. Our results are in accordance with previous studies reporting that intraperitoneal treatment with MaR1 increases adiponectin expression in eWAT of *ob/ob* mice (2 μg/kg; 20 d) and DIO mice (2 μg/kg; 10 d) [12]. Further, MaR1 (35 μg/kg ip; 8 weeks) has also been reported to upregulate circulating levels of adiponectin in DIO mice [14], suggesting the need of higher doses and/or longer periods of treatment with MaR1 for a systemic effect on this adipokine’s levels. Interestingly, literature supports the ability of other SPMs to upregulate *adiponectin* expression in WAT including 17-HDHA [35], RvE1 [36] and RvD1 [37]. Remarkably, the present work shows that MaR1 is able to reverse the decrease in *ADIPOQ* expression induced by HFD in DIO mice and by TNF-α in both models of cultured human adipocytes, which potentially contribute to a parallel improvement on insulin sensitivity or alleviation of hepatic steatosis.

Moreover, MaR1 completely reversed the increase in *leptin* induced by TNF-α in hSP-derived adipocytes, and this effect was mild in hMSC-derived adipocytes. In line with this, Clària et al. [37] reported that treatment with other DHA-derived lipid mediators, RvD1 and RvD2 or a mixture of SPMs (RvD1, RvD2, 17*R*-RvD1, and LXA4), produced either no effects (RvD1) or downregulation (RvD2 or SPMs mixture) of leptin secretion in WAT explants from obese mice. Moreover, regulatory actions on leptin have been previously observed in n-3 PUFA-treated adipocytes. In this case, EPA rather than DHA enhances leptin production in rat primary cultured adipocytes [38] and in 3T3-L1 adipocytes [39]. This may highlight the differential metabolic effects of different SPMs. TNF-α has been reported to regulate leptin production by adipocytes; however, controversial outcomes have been observed. Contrary to other studies reporting that TNF-α inhibits leptin production by adipocytes [40,41], we found that TNF-α enhanced leptin expression after 24 h of exposition in hSP-derived adipocytes, and a similar trend was observed in hMSC-derived adipocytes. One possible explanation is that Fawcett et al. [41] used adipocytes from subcutaneous and omental fat biopsies from morbidly obese subjects (>50 kg/m^2^), while we used subcutaneous fat adipocytes from overweight/obese subjects (<41 kg/m^2^). On the other hand, the study from Yamaguchi et al. [40], was performed in cultured subcutaneous adipocytes from pregnant women. However, in agreement with our results, in vitro and in vivo studies have reported that TNF-α induces leptin expression by adipocytes [42,43,44,45].

We also analyzed the regulatory effects of MaR1 on DPP-4, an enzyme distributed along the body exerting pleiotropic actions on glucose metabolism, insulin sensitivity, gut motility, appetite regulation, inflammation and immune system function through its peptidase activity [46]. DPP-4 was also identified as an adipokine potentially linking obesity and metabolic syndrome [16]. Indeed, *DPP-4* expression in visceral adipose tissue (VAT) has been reported to be increased in overweight and obese subjects and the amount of DPP-4 positively correlated with the amount of VAT, adipocyte size, and adipose tissue inflammation [47]. In contrast, reduced DPP-4 activity has been shown to improve insulin signaling in primary human adipocytes [48]. Moreover, DPP-4 is upregulated by proinflammatory cytokines such as TNF-α in 3T3-L1 [49] and human adipocytes [16]. Surprisingly, our results show that MaR1 increases basal *DPP-4* expression in human subcutaneous adipocytes from overweight/obese subjects. However, it is important to note that MaR1 completely prevented TNF-α-induced upregulation of *DPP-4*, suggesting that MaR1 may be more effective when adipocytes are surrounded by a proinflammatory environment.

Regarding the regulation of *Dpp*-4 in vivo, we did not find any significant change in *Dpp-4* expression in eWAT of DIO mice compared to controls, in concordance with a previous study by our group [12]. In contrast, lower *Dpp-4* expression in WAT depots has been observed in *ob/ob* mice [50], suggesting that other factors different from increased adiposity by itself are involved in the regulation of DPP-4. Even though DPP-4 displays a widespread organ distribution expression, the liver is one of the organs that highly expresses DPP-4 [51]. Most compelling evidence indicates that this enzyme is involved in hepatic damage and steatosis. In this sense, not only *Dpp-4* hepatic mRNA expression has been reported to be upregulated in patients with non-alcoholic fatty liver disease [52], but also associated with steatohepatitis [53]. Our results indicate the MaR1 is able to counteract the HFD-induced increase in hepatic *Dpp-4* mRNA expression, which could also contribute to the antisteatotic actions of MaR1 previously reported by our group [15] and others [14]. Interestingly, a study has proved that hepatocyte-secreted DPP-4 promotes VAT inflammation and insulin resistance in obesity, while silencing the expression of DPP-4 in hepatocytes suppresses both inflammation of VAT and insulin resistance [54]. Notably, we have previously demonstrated the ability of MaR1 to reduce WAT inflammation [12], which could be partially mediated by reduced hepatic *Dpp-4* levels.

Here, we also report for the first time that MaR1 also regulates CT-1, a member of the interleukin (IL)-6 family of cytokines with anti-obesity properties. Indeed, CT-1 deficient mice develop adult-onset obesity, accompanied by insulin resistance and hypercholesterolemia [55]. In contrast, CT-1 administration decreases body weight and fat mass, corrects insulin resistance and liver steatosis in obese mice [55,56]. The present work shows that the expression of *Ct-1* was downregulated in muscle, without significant changes in eWAT of DIO mice. Some authors reported decreased expression of this cytokine in WAT of HFD-fed mice [57,58]. This minor discrepancy in *Ct-1* expression after the HFD-fed period in eWAT could be related to the fact that the HFD period was longer in the study of Pérez-Matute et al. [57] (16 weeks vs. 13 weeks) and to the different obese mouse model being utilized in the study of Sanchez-Infantes et al. [58] (leptin-deficient *ob/ob* mice). Interestingly, our findings show that MaR1 treatment upregulates *Ct-1* expression both in eWAT and muscle. This upregulation of *Ct-1* could also contribute to the beneficial metabolic effects of MaR1 in these tissues [12]. Indeed, CT-1 has been shown to exert potent actions both in WAT and muscle, characterized by an increase in fatty acid oxidation and insulin sensitivity, mediated by the activation of AMPK and AKT respectively [55]. Interestingly, our results also show that MaR1 increases basal *CT-1* expression in both hSP- and hMSC-derived adipocytes. However, contradictory results were detected in the studies of exposure to TNF-α and MaR1 between both models of adipocytes. The highest dose of MaR1 attenuated the potent upregulation of *CT-1* mRNA levels induced by TNF-α in hSP-derived adipocytes, while they were unaffected by TNF-α and increased by MaR1 co-treatment in hMSC-derived adipocytes. This discrepancy in *CT-1* gene expression results may be due to differences in age and BMI of the donors of the adipocytes from hSP and hMSC-derived adipocytes, as well as to differences in culture media composition.

Furthermore, our results show that MaR1 modulates irisin (*FNDC5*) expression in vivo and in vitro. Since the discovery of this myokine and adipokine [59], a growing body of evidence highlights its potential therapeutic aspects including anti-inflammatory effects [60], promotion of WAT browning [59], enhancement of lipolysis-related genes, decreasing lipid accumulation [61], as well as increasing glucose uptake by beige adipocytes [62]. In fact, Boström et al. [59] demonstrated that adenovirus-mediated moderate augmentation in circulating irisin levels increased energy expenditure, reduced body weight and improved diet-induced insulin resistance.

Irisin has been observed to possess anti-inflammatory properties by reducing the expression and activity of major proinflammatory cytokines including TNF-α in 3T3-L1 adipocytes [60]. To our knowledge this is the first report showing the effects of MaR1 and TNF-α on *FNDC5* expression by human adipocytes. However, we observed contradictory results for the gene expression levels of *FNDC5* in basal and TNF-α conditions between hMSC and hSP-derived adipocytes. In basal conditions, MaR1 upregulated gene expression of *FNDC5* in hMSC-derived adipocytes, while no effect was found in hSP-derived adipocytes. In addition, *FNDC5* mRNA levels were not changed by TNF-α in hMSC-derived adipocytes, but increased by TNF-α in hSP-derived adipocytes, and the effect was reversed by MaR1 at the highest dose. On the contrary, when hMSC-derived adipocytes were treated with TNF-α and the highest dose of MaR1, gene expression of *FNDC5* was significantly enhanced. Again, differences between the adipocytes’ origin and differentiation protocol might account for the differential outcomes.

In addition, our in vivo results showed an increased *Fndc5* expression in eWAT of DIO according to previous studies [63]. Contrariwise, other investigations reported no differences [64,65] or even a fall in WAT *Fndc5* expression [66]. Irisin is mainly produced by muscle cells although small amounts of this peptide have been detected in WAT, brain, subcutaneous glands, liver, stomach, spleen and testis [67]. The current data show a decreased expression of *Fndc5* expression in skeletal muscle of DIO mice, in line with other studies [68,69]. Given that muscle produces the highest amount of this peptide, a decreased muscle production of irisin may have a great impact on circulating levels. In this sense, lower circulating irisin levels have been reported in DIO mice [70,71] and also in diabetic *db/db* mice [68]. Thus, it is important to highlight the ability of MaR1 to upregulate muscle *Fndc5* expression. Interestingly, a similar effect on *Fndc5* muscle expression has been found after diet supplementation with the antidiabetic agent metformin in diabetic *db/db* mice [68]. These results may suggest that the beneficial actions of MaR1 on insulin sensitivity could be partially mediated by the regulation of muscle irisin production.

On the other hand, it should be mentioned that effects of MaR1 and TNF-α, alone or in combination, on adipokine secretion were usually more moderate than those observed on adipokine expression in both models of human adipocytes. This apparent discrepancy between expression and secretion data could be due to the fact that cultured medium for the secretion experiments were collected after 24 h treatments and the adipokine secretion is continuous and cumulative. Probably, longer MaR1 treatment (48–72 h) would likely be needed to obtain adipokine secretion levels that reflect the expression at 24 h. However, with our current data we can not rule out than other post-transcriptional mechanisms could be involved in the regulation of adipokine secretion by both MaR1 and TNF-α.

Another interesting issue to be also addressed in future studies is to characterize the mechanisms by which MaR1 is able to regulate basal secretion of the studied adipokines and the pathways involved in its preventive actions of the alterations induced by TNF-α on adipokine expression. In this way, previous studies of our group have shown that MaR1 is able to counteract the upregulation of pERK/ERK ratio induced by TNF-α in 3T3-L1 adipocytes [25]. Interestingly, MaR1 also prevented the inhibitory effect of TNF-α on insulin-stimulated Akt phosphorylation in hMSC-adipocytes [24]. It would be also of interest to characterize if MaR1 actions on adipokine expression are mediated through the activation of the LGR6 receptor, which has been identified as the receptor mediating MaR1 actions in different cell types [72].

Although some of our current findings about MaR1 effects on adipokines in cultured adipocytes mimic the observations found in WAT of DIO mice (i.e., for adiponectin), an apparent discrepancy between the in vitro and in vivo results was observed for other adipokines (i.e., *leptin*, *Fndc5*). However, it is important to take into account that the in vitro cultured adipocytes only evaluate the direct effects of MaR1 treatment on adipokine production, while in the in vivo models the expression of adipokines in WAT could be influenced by other factors/mechanisms that are not present in the in vitro system (cellular heterogeneity, inter-organ crosstalk and the differential regulation in each tissue suggested by our results, which may lead to the activation of regulatory/compensatory mechanisms). Moreover, it should be also considered a potential differential response between species (human adipocytes from obese/overweight subjects used in our in vitro studies, while the in vivo experiments were performed in DIO mice). Additionally, it would be interesting to assess the effects of MaR1 on adipocytes derived from different human fat depots, based on the metabolic and functional heterogeneity described for subcutaneous and visceral adipose tissue, the latter being more prone to inflammation [6]. On the other hand, the lack of trials assaying the effects of MaR1 in human subjects limits the comparability of the results. However, the fact that MaR1 regulates adipokine expression in human adipocytes leads us to suggest a potential physiological role of this SPM in human obesity in vivo, highlighting the relevance of future research characterizing changes in MaR1 levels in WAT of obese subjects who are apparently healthy or with metabolic complications, such as T2DM.

Importantly, our data support the relevance of testing the effects of bioactive molecules in different models of adipocytes in order to avoid bias and to strengthen the translational value of the results.

## 5. Conclusions

In summary, data of the current study demonstrate that MaR1 regulates the expression pattern of adipokines in human adipocytes towards a healthier metabolic state. Similar results were found in both models of cultured human adipocytes, although with some discrepancy possibly due to differences in adipocyte characteristics and differentiation procedures. Of note, the effect of TNF-α on hMSC-derived adipocytes seems to be weaker than on hSP-derived adipocytes for all the genes analyzed, and some of the effects of MaR1 were specific to hMSC or hSP. In addition, we have described the differential tissue regulation of these bioactive proteins by obesity and MaR1 treatment in WAT, liver and skeletal muscle. Overall, our data suggest that the beneficial effects of MaR1 on insulin sensitivity and liver steatosis might also rely on its ability to control the secretome of eWAT, liver and muscle, and therefore the crosstalk between key metabolic organs.

## Figures and Tables

**Figure 1 biomolecules-13-00919-f001:**
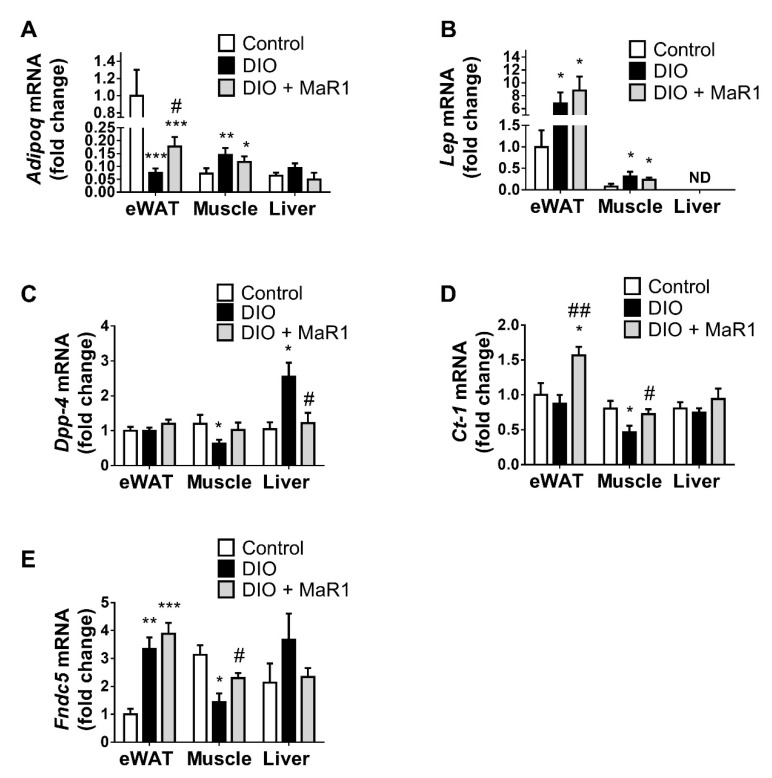
In vivo treatment with MaR1 regulates the expression of adipokines in DIO mice. mRNA levels of (**A**) *Adiponectin* (*AdipoQ*), (**B**) *Leptin* (*Lep*), (**C**) *Dipeptidyl peptidase-4* (*Dpp-4*), (**D**) *Cardiothrophin-1 (Ct-1*), and (**E**) *Irisin* (*Fndc5*) in eWAT, soleus muscle and liver in control and DIO mice treated by oral gavage with either vehicle or MaR1 (50 μg/kg, 10 d; *n* = 7–8 animals per group). Data were analyzed by one-way ANOVA and expressed as mean ± SEM. * *p* < 0.05; ** *p* < 0.01; *** *p* < 0.001 vs. control mice; # *p* < 0.05; ## *p* < 0.01 vs. DIO mice.

**Figure 2 biomolecules-13-00919-f002:**
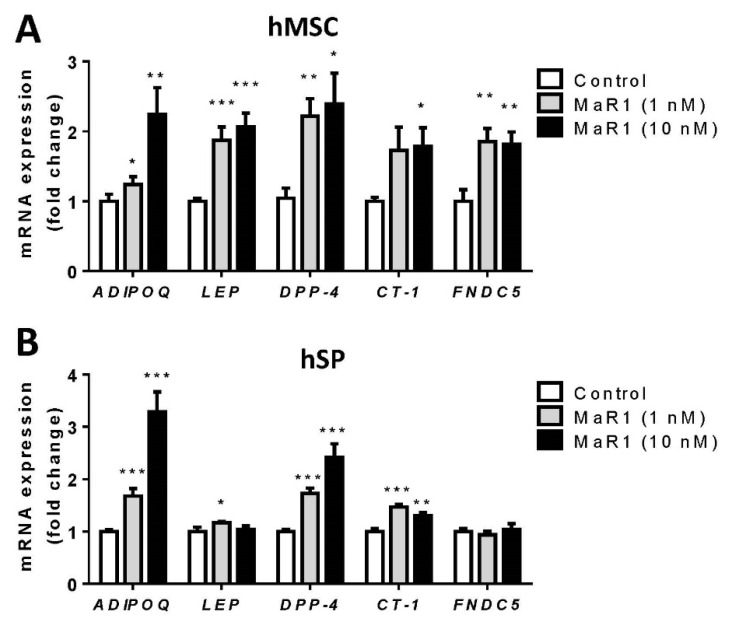
MaR1 regulates adipokine expression in hMSC-derived and hSP-derived adipocytes. Gene expression of Adiponectin (*ADIPOQ*), Leptin (*LEP*), Dipeptidyl peptidase-4 (*DPP-4*), Cardiotrophin-1 (*CT-1*) and Irisin (*FNDC5*) in (**A**) hMSC-derived adipocytes (hMSC) and (**B**) hSP-derived adipocytes (hSP), treated with MaR1 (1 and 10 nM) for 24 h. Data were analyzed by one-way ANOVA and expressed as mean ± SEM. (*n* = 5–9) from different sets of experiments. * *p* < 0.05; ** *p* < 0.01; *** *p* < 0.01 vs. control vehicle-treated cells.

**Figure 3 biomolecules-13-00919-f003:**
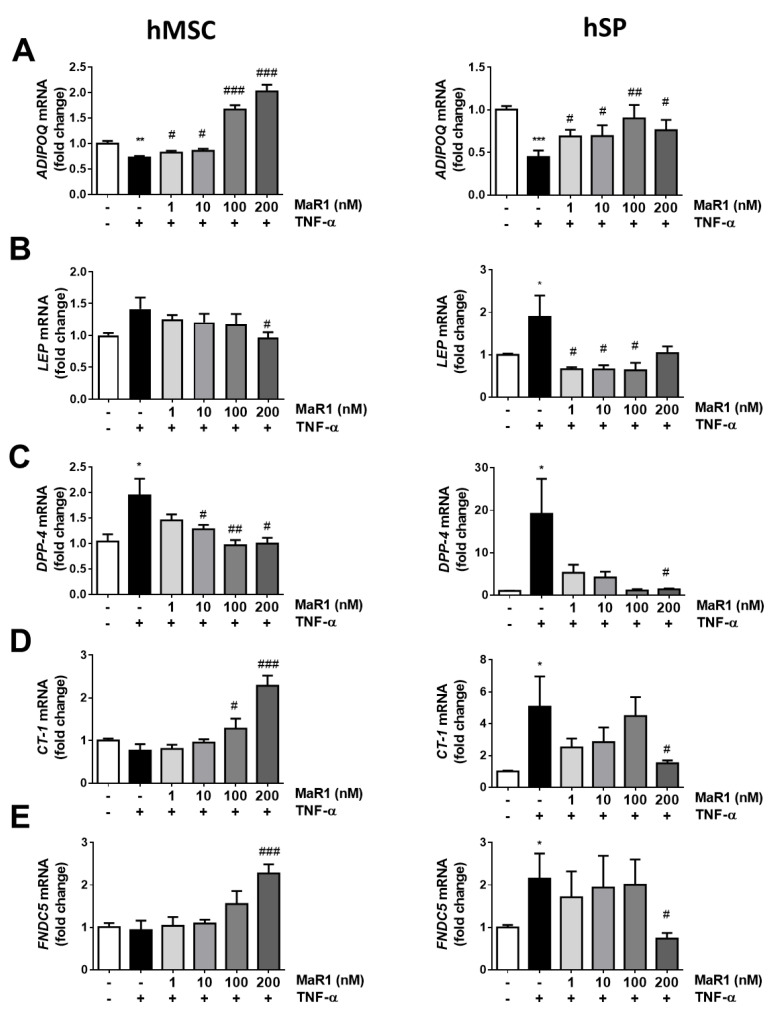
MaR1 modulates the effects of TNF-α on adipokines’ expression in hMSC-derived adipocytes and hSP-derived adipocytes. Gene expression of (**A**) Adiponectin (*ADIPOQ*), (**B**) Leptin (*LEP*), (**C**) Dipeptidyl peptidase-4 (*DPP-4*), (**D**) Cardiotrophin-1 (*CT-1*) and (**E**) *FNDC5* in hMSC-derived (left panels) and hSP-derived (right panels) adipocytes treated for 24 h with TNF-α (100 ng/mL) in the absence or presence of MaR1 (1–200 nM). Data were analyzed by one-way ANOVA and expressed as mean ± SEM. (n = 4–9 from different sets of experiments). * *p* < 0.05; ** *p* < 0.01; *** *p* < 0.001 vs. control vehicle-treated cells; # *p* < 0.05; ## *p* < 0.01; ### *p* < 0.001 vs. TNF-α treated cells.

## Data Availability

Not applicable.

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
