# Peer review of "Maresin 1 Exerts a Tissue-Specific Regulation of Adipo-Hepato-Myokines in Diet-Induced Obese Mice and Modulates Adipokine Expression in Cultured Human Adipocytes in Basal and Inflammatory Conditions"

_biomolecules, 2023, doi:10.3390/biom13060919_

Round 1
Reviewer 1 Report
In the present manuscript, Martinez et al. studied the effect of Maresin 1 (MaR1), a DHA-derived specialized proresolving lipid mediator, on the expression of adipokines including adiponectin, leptin, dipeptidyl peptidase 4 (DPP-4), cardiotrophin-1 (CT-1), and irisin (FNDC5) in both in vitro and in vivo models of obesity. They observed that MaR1 treatment in diet-induced obese mice led to increased expression of adiponectin and CT-1 in epididymal adipose tissue, increased expression of FNDC5 and CT-1 in muscle, and decreased expression of DPP-4 in liver. In human differentiated primary adipocytes, MaR1 increased the expression of adiponectin, leptin, DPP4, CT-1 and FNDC5. MaR1 also counteracted the TNF-α-induced downregulation of adiponectin and the upregulation of DPP-4 and leptin observed in adipocytes. Given that the number of research studies on the effects of MaR1 on adipokines secreted by adipocytes is still relatively limited, studies such as the present report are timely. However, the current study in the present format provides a limited knowledge on the effect of MaR1 on adipokines production since the effect of MaR1 is only studied and gene expression level. Below are my major points or concerns:
1. Authors have concluded in different occasions throughout the manuscript (see examples below) that they have showed that MaR1 have the ability to regulate the secretion or production of the adipokines.
The current study is somewhat limited as authors only measure mRNA expression levels and not protein levels. Increased mRNA levels do not necessary translate into increased secretion of corresponding proteins mainly due to various post-transcriptional (let’s not to forget about post-translational regulatory mechanisms that might be less important here) that might be in place. To me, authors need to confirm their major findings by measuring protein levels of the adipokines, etc by methods such as ELISA, or proteomics, or similar.
Example 1 : we proved the ability of this SPM to regulate the adipokines production…..
Example 2: In summary, data of the current study demonstrate that MaR1 regulates the secretory pattern of adipokines in human adipocytes towards a healthier metabolic state.
2. It would be useful for the readers if authors explain the reason, they chose oral vs. intraperitoneal injection of MaR1 and if they expect any differences in the result outcome by choosing any of these two administration routs.
3. For statistical analysis, authors have mentioned that they used various statistical tests. However, they need to specify for each figure that which test was used for that specific figure.
4. MaR1, DHA and TNF are not defined for their abbreviation in the abstract.
5. The manuscript will benefit from some language editing/spelling checks. Below are some examples:
Example 1: MaR1 reverses in a tissue-dependent manner the expression of specific adipomyokines and hepatokines altered by obesity in mice.
Suggestion: MaR1 reverses the expression of specific adipomyokines and hepatokines altered in obese mice in a tissue-dependent manner…….
Example 2: A previous study of our group
Suggestion: A previous study by (or done, or from) our group
Example 3: Other highly-studied hormone
Suggestion: Other well-studied hormone
Example 4: Therefore, the aim of the present study was to analyze the efficacy of MaR1 to reverse in a tissue-dependent manner the changes induced by obesity on the expression of these adipokines in WAT, liver and muscle of DIO mice
Suggestion: Therefore, the aim of the present study was to analyze the efficacy of MaR1 to reverse changes induced by obesity on the expression of these adipokines in WAT, liver and muscle of DIO mice in a tissue-dependent manner.
Example 5: However, accordingly to our results, in vitro and in vivo studies have reported that TNF-α induces leptin production by adipocytes.
Suggestion: However, in line with (in agreement with) our results, in vitro and in vivo studies have reported that TNF-α induces leptin production by adipocytes.
Author Response
Reviewer #1
Comments and Suggestions for Authors
In the present manuscript, Martinez et al. studied the effect of Maresin 1 (MaR1), a DHA-derived specialized proresolving lipid mediator, on the expression of adipokines including adiponectin, leptin, dipeptidyl peptidase 4 (DPP-4), cardiotrophin-1 (CT-1), and irisin (FNDC5) in both in vitro and in vivo models of obesity. They observed that MaR1 treatment in diet-induced obese mice led to increased expression of adiponectin and CT-1 in epididymal adipose tissue, increased expression of FNDC5 and CT-1 in muscle, and decreased expression of DPP-4 in liver. In human differentiated primary adipocytes, MaR1 increased the expression of adiponectin, leptin, DPP4, CT-1 and FNDC5. MaR1 also counteracted the TNF-α-induced downregulation of adiponectin and the upregulation of DPP-4 and leptin observed in adipocytes. Given that the number of research studies on the effects of MaR1 on adipokines secreted by adipocytes is still relatively limited, studies such as the present report are timely. However, the current study in the present format provides a limited knowledge on the effect of MaR1 on adipokines production since the effect of MaR1 is only studied and gene expression level. Below are my major points or concerns:
- Authors have concluded in different occasions throughout the manuscript (see examples below) that they have showed that MaR1 have the ability to regulate the secretion or production of the adipokines.
The current study is somewhat limited as authors only measure mRNA expression levels and not protein levels. Increased mRNA levels do not necessary translate into increased secretion of corresponding proteins mainly due to various post-transcriptional (let’s not to forget about post-translational regulatory mechanisms that might be less important here) that might be in place. To me, authors need to confirm their major findings by measuring protein levels of the adipokines, etc by methods such as ELISA, or proteomics, or similar.
Example 1 : we proved the ability of this SPM to regulate the adipokines production…..
Example 2: In summary, data of the current study demonstrate that MaR1 regulates the secretory pattern of adipokines in human adipocytes towards a healthier metabolic state.
Response: We thank the reviewer for his/her suggestion. As we totally agree with this comment, we performed new ELISA experiments to analyze the effects of MaR1 and TNF-alpha alone or in combination on the secretion of the different adipokines. We were able to run these experiments since we had aliquots of media at -80ºC collected after 24 h of treatment from the cultures in which we measured mRNA expression.
We have included these new data in the new Supplemental Figures 1 and 2, and we have described the results obtained in the revised version of the manuscript (the new text is highlighted in red).
It should be mentioned that the effects observed on adipokine secretion were more moderate than those observed on mRNA expression and not always follow the same pattern. This issue has been discussed in a new paragraph of the revised version of the manuscript (see lines 479-488 of the revised manuscript).
Moreover, considering that that the main effects were observed on mRNA expression, we have followed the suggestion of the reviewer and we have gone through the text to change “expression” for “secretion” were convenient.
All the comparative studies on the effects of MaR1 adipokine’s mRNA expression between hMSC-derived adipocytes and hSP-derived adipocytes (previously showed in Supplemental data) are now shown together in the new version of Figures 2 and 3 of the revised manuscript.
- It would be useful for the readers if authors explain the reason, they chose oral vs. intraperitoneal injection of MaR1 and if they expect any differences in the result outcome by choosing any of these two administration routs.
Response: We thank the reviewer for his/her comment. In previous studies from our group, we tested the effectiveness of both oral and intraperitoneal injection of MaR1. Our data showed that oral administration of Maresin 1 to mice is also effective as intraperitoneal injection to treat obesity-related disorders (Laiglesia et al 2017, “Maresin 1 mitigates liver steatosis in ob/ob and diet-induced obese mice”). This effect is relevant for the potential therapeutic use of Maresin 1 in humans.
- For statistical analysis, authors have mentioned that they used various statistical tests. However, they need to specify for each figure that which test was used for that specific figure.
Response: We thank the reviewer for his/her suggestion. The statistical test is specified in its respective figure legend.
- MaR1, DHA and TNF are not defined for their abbreviation in the abstract.
Response: We thank the reviewer for his/her suggestion. We have defined it in the abstract of the revised manuscript.
- The manuscript will benefit from some language editing/spelling checks. Below are some examples:
Example 1: MaR1 reverses in a tissue-dependent manner the expression of specific adipomyokines and hepatokines altered by obesity in mice.
Suggestion: MaR1 reverses the expression of specific adipomyokines and hepatokines altered in obese mice in a tissue-dependent manner…….
Example 2: A previous study of our group
Suggestion: A previous study by (or done, or from) our group
Example 3: Other highly-studied hormone
Suggestion: Other well-studied hormone
Example 4: Therefore, the aim of the present study was to analyze the efficacy of MaR1 to reverse in a tissue-dependent manner the changes induced by obesity on the expression of these adipokines in WAT, liver and muscle of DIO mice
Suggestion: Therefore, the aim of the present study was to analyze the efficacy of MaR1 to reverse changes induced by obesity on the expression of these adipokines in WAT, liver and muscle of DIO mice in a tissue-dependent manner.
Example 5: However, accordingly to our results, in vitro and in vivo studies have reported that TNF-α induces leptin production by adipocytes.
Suggestion: However, in line with (in agreement with) our results, in vitro and in vivo studies have reported that TNF-α induces leptin production by adipocytes.
Response: We thank the reviewer for all these helpful comments. We changed those examples and checked for similar ones in the revised version of the manuscript (changes performed y are now highlighted in red in the text).
Reviewer 2 Report
The manuscript by Martínez-Fernández et al. highlights the role Maresin-1, an anti-inflammatory mediator, in the regulation of different cytokines at different mice tissues. Also, the authors used adipocytes mediated hMSCs and hAd-MSCs to replicate their observation in human. TNFa – mediated inflammation was normalized post co-treatment with Maresin-1, indicating a prospective role of the later in regulating TNFa pathway. The study is will designed but I have the following concerns:
1. The study is based on observations for the role of Merasin-1. No mechanism was provided. The prospective role of TNFa pathways implication should be further studied to delineate the role of Maresin-1 in this pathway.
2. The gene regulation was observed at mRNA level of the studied gene. It would be ideal to study the gene regulation at the protein levels, e.g. Western Blot and/or Immunofluoresce assays.
3. The Manuscript need an English language edit.
Author Response
Comments and Suggestions for Authors
The manuscript by Martínez-Fernández et al. highlights the role Maresin-1, an anti-inflammatory mediator, in the regulation of different cytokines at different mice tissues. Also, the authors used adipocytes mediated hMSCs and hAd-MSCs to replicate their observation in human. TNFa – mediated inflammation was normalized post co-treatment with Maresin-1, indicating a prospective role of the later in regulating TNFa pathway. The study is will designed but I have the following concerns:
- The study is based on observations for the role of Merasin-1. No mechanism was provided. The prospective role of TNFa pathways implication should be further studied to delineate the role of Maresin-1 in this pathway.
Response: We thank the reviewer for this comment. We agree with the reviewer about the interest of characterizing the mechanisms by which MaR1 regulates both basal adipokine’s expression and the alterations induced by TNF-α. This issue has been now mentioned and discussed in a new paragraph of the revised version of the manuscript (see lines 489-498), that reads as follow: “Another interesting issue to be also addressed in future studies will be to characterize the mechanisms by which MaR1 is able to regulate basal secretion of the studied adipokines and the pathways involved in its preventive actions of the alterations induced by TNF-α on adipokine’s expression. In this way, previous studies of our group have shown that MaR1 is able to counteract the upregulation of pERK/ERK ratio induced by TNF-α in 3T3-L1 adipocytes (Laiglesia et al., 2018). Interestingly, MaR1 also prevented the inhibitory effect of TNF-α on insulin-stimulated Akt phosphorylation in hMSC-adipocytes (Martínez-Fernández et al., 2021). It would be also of interest to characterize if MaR1 actions on adipokine’s expression are mediated through the activation of the LGR6 receptor, which has been identified as the receptor mediating MaR1 actions in different cell types (Chiang et al., 2019)”
- The gene regulation was observed at mRNA level of the studied gene. It would be ideal to study the gene regulation at the protein levels, e.g. Western Blot and/or Immunofluoresce assays.
Response: We agree with this helpful suggestion of the reviewer. Accordingly, we performed ELISA experiments to analyze the secretion of the adipokines in both models of human adipocytes after the treatment for 24 h with MaR1, TNF-alpha or a combination of both.
We have included these new data in the new Supplemental Figures 1 and 2, and we have described the results obtained in the revised version of the manuscript (the new text is highlighted in red). The effects observed on adipokine secretion were more moderate than those observed on mRNA expression and not always follow the same pattern. This issue has been discussed in a new paragraph of the revised version of the manuscript (see lines 479-488 of the revised manuscript).
All the comparative studies on the effects of MaR1 adipokine’s mRNA expression between hMSC-derived adipocytes and hSP-derived adipocytes are now shown together in the new version of Figures 2 and 3 of the revised manuscript.
- The Manuscript need an English language edit.
Response: We thank the reviewer for this suggestion. We made changes in the text to improve the English language which are highlighted it the text.
Round 2
Reviewer 1 Report
It's great to see that authors were able to add measures of protein levels through ELISA and also improve the text. These additions will undoubtedly enhance the overall quality and impact of their work.
However, they are few minor suggestions that still need to be addressed:
1. As the authors pointed out, the effects observed on adipokine secretion were more moderate than those observed on mRNA expression and did not always follow the same pattern. There are places in the text where this distinction is not accurately reflected. For example, in line 278-281, the tone of the sentence could be revised to better align with the ELISA data. Perhaps something like: "Similar to what was observed on ADIPOQ mRNA expression, MaR1 treatment induced a moderate but statistically significant increase in adiponectin secretion in hMSC-derived adipocytes, in contrast to hSP-derived adipocytes (Figure S1 A and B, respectively)."
2. Authors have appropriately used "One-way ANOVA" for statistical analysis of their data. This is an appropriate test since more than 2 groups are compared. However in Supplementary Figure 1, authors instead used Student t-test or U-Mann-Whitney which are not recommended when more than 2 groups are being compared. Authors have not provided any explanation why they used Student t-test or U-Mann-Whitney (recommended for comparing two groups) instead of One-way ANOVA.
3. The manuscript would benefit from language editing to address the spelling errors and improve the overall readability. For instance, in line 279, "expresion" should be corrected to "expression". There may be more instances like this that could be corrected through a thorough review.
Author Response
It's great to see that authors were able to add measures of protein levels through ELISA and also improve the text. These additions will undoubtedly enhance the overall quality and impact of their work.
However, they are few minor suggestions that still need to be addressed:
- As the authors pointed out, the effects observed on adipokine secretion were more moderate than those observed on mRNA expression and did not always follow the same pattern. There are places in the text where this distinction is not accurately reflected. For example, in line 278-281, the tone of the sentence could be revised to better align with the ELISA data. Perhaps something like: "Similar to what was observed on ADIPOQ mRNA expression, MaR1 treatment induced a moderate but statistically significant increase in adiponectin secretion in hMSC-derived adipocytes, in contrast to hSP-derived adipocytes (Figure S1 A and B, respectively)."
Response: We thank the reviewer for this comment. As we agree with the comment and suggestion, we have included the suggested sentence (changes performed are now highlighted in red in the revised manuscript.
- Authors have appropriately used "One-way ANOVA" for statistical analysis of their data. This is an appropriate test since more than 2 groups are compared. However in Supplementary Figure 1, authors instead used Student t-test or U-Mann-Whitney which are not recommended when more than 2 groups are being compared. Authors have not provided any explanation why they used Student t-test or U-Mann-Whitney (recommended for comparing two groups) instead of One-way ANOVA.
Response: We totally agree with the reviewer that One-way ANOVA is the appropriate test when more than 2 groups are compared. In the case of Supplementary Figure 1 we used Student t-test or U-Mann-Whitney since in some adipocyte cultures we have only secretion data for one dose of MaR1 (1 or 10 nM). For this reason, we decided to compare the effect of each dose of MaR1 with the control, and not to compare differences between doses of MaR1.
- The manuscript would benefit from language editing to address the spelling errors and improve the overall readability. For instance, in line 279, "expresion" should be corrected to "expression". There may be more instances like this that could be corrected through a thorough review
Response: We thank the reviewer for this comment. We apologize for the spelling errors. The manuscript has been edited to address the spelling errors and improve the overall readability